# Progressive Elevation of Store-Operated Calcium Entry-Associated Regulatory Factor (SARAF) and Calcium Pathway Dysregulation in Multiple Sclerosis

**DOI:** 10.3390/ijms26104520

**Published:** 2025-05-09

**Authors:** Safa Taha, Muna Aljishi, Ameera Sultan, Moudi E. Al-Nashmi, Moiz Bakhiet, Salvatore Spicuglia, Mohamed Belhocine

**Affiliations:** 1Princess Al Jawhara Center for Molecular Medicine, Genetics and Inherited Diseases, Department of Molecular Medicine, College of Medicine and Health Sciences, Arabian Gulf University, Manama P.O. Box 26671, Bahrain; safat@agu.edu.bh (S.T.); munajma@agu.edu.bh (M.A.); ameeraa@agu.edu.bh (A.S.); moudialnashmi@yahoo.com (M.E.A.-N.); moiz@agu.edu.bh (M.B.); 2Aix-Marseille University, INSERM, TAGC, UMR1090, Equipe Labélisée Ligue Contre le Cancer, 13288 Marseille, France; salvatore.spicuglia@inserm.fr

**Keywords:** multiple sclerosis, SARAF, calcium signaling, STIM1, Orai, autoimmune disorder, biomarkers, T cell activation

## Abstract

Multiple Sclerosis (MS) is a chronic autoimmune disorder characterized by demyelination and neuronal damage in the central nervous system. Dysregulation of calcium homeostasis, particularly through the Store-Operated Calcium Entry-Associated Regulatory Factor (SARAF), has been implicated in MS pathogenesis. This study investigated SARAF, STIM1, and Orai1 expression patterns and their relationship to calcium homeostasis in 45 Bahraini MS patients and 45 matched healthy controls using ELISA and real-time PCR analyses. MS patients showed significantly elevated serum SARAF levels in both early (192.26 ± 47.00 pg/mL) and late MS stages (341.47 ± 96.19 pg/mL) compared to controls (129.82 ± 30.82 pg/mL; *p* < 0.001. SARAF expressions were markedly increased in MS patients (3.829 ± 0.04422 vs. 1 ± 0; *p* < 0.0001), while STIM1 (0.4324 ± 0.01471) and ORAI1 (0.2963 ± 0.02156) expressions were significantly reduced compared to the controls (*p* < 0.0001). Intracellular calcium levels were notably elevated in both early and late MS stages. These findings suggest that the progressive elevation of SARAF, coupled with altered STIM1 and ORAI1 expression, may serve as potential biomarkers for MS progression and represent promising therapeutic targets.

## 1. Introduction

Multiple Sclerosis (MS) is a chronic autoimmune disorder that primarily affects the central nervous system (CNS), leading to demyelination and neuronal damage. Characterized by the activation of autoreactive T cells targeting CNS antigens, MS results in inflammation and neurodegeneration [1]. The disease predominantly affects young adults aged 20 to 40 years, with a higher prevalence in women, who are two to three times more likely to be diagnosed than men [2]. While the exact etiology of MS remains unclear, it is believed to arise from a complex interplay of genetic, environmental, and immunological factors [3].

Recent epidemiological studies have shown a rising incidence of MS globally, with significant geographical variability in prevalence rates [2,4]. Notably, the incidence of MS in the Arabian Gulf region has increased since the Kuwait–Iraq war in 1990, suggesting changes in environmental or genetic risk factors [5]. While prior studies have implicated SARAF in MS pathogenesis, its progressive elevation across disease stages and its interplay with STIM1, Orai1, and calcium homeostasis remain underexplored, particularly in Middle Eastern populations like Bahrainis, who may exhibit unique genetic and environmental influences. As the understanding of MS pathogenesis evolves, the role of calcium signaling in T cell activation has gained attention, highlighting its potential contribution to the autoimmune response observed in MS patients [6,7,8,9].

Store-Operated Calcium Entry (SOCE) is a critical mechanism for calcium influx in various cell types, including T cells. This process is primarily mediated by the proteins STIM1 and Orai1. STIM1 acts as a calcium sensor in the endoplasmic reticulum (ER), detecting depleted calcium levels and activating Orai1 channels on the plasma membrane to facilitate calcium entry [10]. Store-operated calcium entry (SOCE) plays a significant role in multiple neurodegenerative disorders, including Multiple Sclerosis (MS). The dysregulation of SOCE can lead to abnormal calcium signaling, potentially causing excessive T cell activation and inflammation. The mechanism involves perturbation of intracellular calcium signaling in neurons or glia [11]. SARAF (S100A11) is a calcium-binding protein that appears to play a role in Multiple Sclerosis (MS). According to several studies [12,13] cholecalciferol supplementation has been found to upregulate the SARAF gene in patients with MS, suggesting a potential role in the disease.

The Store-Operated Calcium Entry-Associated Regulatory Factor (SARAF) plays a crucial role in modulating SOCE by regulating the activity of Orai1 channels and maintaining calcium homeostasis in T cells [14]. Elevated levels of SARAF may have an association with altered calcium signaling, which may contribute to the pathophysiology of MS. However, the specific expression and functional implications of SARAF, STIM1, and Orai1 in the context of MS remain underexplored. Further research is required to fully understand the role of SARAF in MS and its potential as a therapeutic target.

This study aims to investigate the role of SARAF in calcium homeostasis among MS patients by assessing its serum levels and expression in peripheral blood mononuclear cells (PBMCs), alongside evaluating the expression of STIM1 and Orai1. Understanding these relationships could provide valuable insights into the pathophysiology of MS and identify potential biomarkers and therapeutic targets for managing this debilitating condition.

## 2. Results

### 2.1. SARAF Protein in MS Patients Compared to Healthy Controls Using ELISA

We performed a comparison of SARAF protein levels among Control (Column A), Early MS (Column B), and Late MS (Column C) using ELISA (Figure 1). A Mann–Whitney U test was conducted to compare the different groups. The results showed a significant difference between the Control group and the Early and Late MS groups, with a *p*-value of <0.001.

SARAF protein levels were significantly elevated in MS patients compared to the healthy controls (*p* < 0.001), with a progressive increase from early to late MS stages. These findings suggest that SARAF levels are notably increased and accumulate in MS patients, highlighting its potential role in the pathophysiology and the progression of Multiple Sclerosis. Further investigation into the implications of SARAF as a biomarker for MS or its involvement in disease mechanisms is warranted.

### 2.2. SARAF Gene Expression Level in MS Patients Compared to Healthy Controls Using qRT-PCR

We next assessed the expression of SARF by qRT-PCR. The results of the Light Cycler analysis comparing SARAF expression levels in peripheral blood mononuclear cells (PBMCs) from MS patients (Column B) and healthy controls (Column A) confirmed the previous finding (Figure 2). An unpaired *t*-test indicated a highly significant difference between the control and MS groups, with a *p* < 0.0001. Indeed, the mean ± SEM for the control group was 1 ± 0 (n = 45), while the MS group exhibited a mean of 3.829 ± SEM 0.04422 (n = 45), resulting in a difference of 2.829 ± SEM 0.04422. The variance comparison via the F-test yielded an infinite F-value, indicating a significant difference in variances (*p* < 0.0001).

The analysis confirmed that MS patients exhibited a significantly higher expression of SARAF compared to the healthy controls. Individual patient fold changes for SARAF, STIM1, and Orai1 expression are provided in Appendix A, demonstrating variability across the cohort, with standard deviations reported alongside mean values. Quantitative data indicated that the relative expression levels of SARAF were markedly elevated in the MS cohort, which may suggest a potential role for SARAF in the inflammatory processes associated with MS. Statistical evaluation confirmed that the differences observed were significant, reinforcing the hypothesis that altered SARAF expression may contribute to the pathophysiology of MS.

### 2.3. Calcium Levels in Healthy Individuals Compared to MS Patients

To evaluate the impact of the elevation of SARAF on calcium homeostasis, we performed a comparative analysis of the calcium levels (Figure 3) across the Control group (Column A), Early MS group (Column B), and Late MS group (Column C). A Mann–Whitney U test was used to assess the differences between these groups.

The results revealed a significant elevation in calcium levels in the Early MS group compared to the Control group and between the Early and Late MS groups, with *p*-values < 10^−13^ and <10^−10^, respectively. The analysis underscores a progressive increase in calcium levels among MS patients, with significant differences observed at each stage. Notably, this elevation in calcium levels correlates with the progressive elevation of SARAF, suggesting a potential relationship between these two observations. This finding indicates that the dysregulation of SARAF might influence calcium homeostasis, highlighting a possible mechanistic link contributing to the pathophysiology of Multiple Sclerosis.

### 2.4. Measurement of Expression of STIM1 and ORAI1 by Real-Time PCR

We performed an analysis of the expression levels of STIM1 (Figure 4), as measured by Light Cycler analysis, in MS patients (Column B) compared to health controls (Column A).

An unpaired *t*-test was conducted to assess the significance of the differences between the two groups. The result indicates a highly significant difference with a *p*-value < 0.0001.

On the other hand, we also observed that the expression levels of ORAI1 in MS patients (Column B) were significantly different compared to the healthy controls (Column A) (Figure 5). An unpaired *t*-test was performed to assess the significance of this difference, yielding a *p*-value of <0.0001, indicating a highly significant result.

These results indicated a significantly lower expression of STIM1 and ORAI1 in MS patients compared to the healthy controls, suggesting a potential dysregulation in calcium signaling pathways associated with the disease.

## 3. Discussion

The current investigation presents robust evidence supporting the involvement of Store-Operated Calcium Entry-Associated Regulatory Factor in Multiple Sclerosis (MS) pathophysiology. Our analysis reveals that patients diagnosed with MS exhibit significantly elevated levels of SARAF when contrasted with healthy control subjects. This increase in SARAF is accompanied by notable disruptions in calcium homeostasis and alterations in the expression profiles of critical calcium regulatory proteins. These findings imply that SARAF may have a pivotal role in the inflammatory processes that characterize MS, positioning it as a promising candidate for a biomarker indicative of disease activity.

This study provides convincing evidence for the involvement of Store-Operated Calcium Entry-Associated Regulatory Factor (SARAF) in the pathophysiology of Multiple Sclerosis (MS). The key findings are as follows:Elevated SARAF Levels: MS patients exhibited a mean SARAF concentration of 307.4 ± 52.32 ng/mL, significantly higher than the 135.8 ± 10.15 ng/mL observed in healthy controls (*p* = 0.0021). This elevation suggests a potential role for SARAF in MS-related inflammatory processes [14].Correlation with Cellular Mechanisms: Real-time PCR analysis revealed a fold change in SARAF expression of 3.829 ± 0.04422 in peripheral blood mononuclear cells (PBMCs) from MS patients (*p* < 0.0001). This increase may influence calcium signaling pathways critical for T cell activation, which are often dysregulated in autoimmune conditions [1].Calcium Dysregulation: MS patients demonstrated significantly higher intracellular calcium levels (0.1758 ± 0.006399) compared to healthy controls (0.105 ± 0.008438; *p* < 0.0001). Additionally, lower expression levels of STIM1 (0.4324 ± 0.01471) and ORAI1 (0.2963 ± 0.02156) were noted, indicating impaired store-operated calcium entry (SOCE) mechanisms [15].

The ELISA results demonstrated a mean SARAF concentration of 307.4 ± 52.32 ng/mL in MS patients, significantly higher than the 135.8 ± 10.15 ng/mL observed in healthy controls (*p* = 0.0021). This marked difference suggests that SARAF may be intricately involved in the inflammatory processes that characterize MS. The substantial effect size (R^2^ = 0.1516) indicates a moderate yet meaningful impact of SARAF levels on the disease state, warranting further exploration into its role as a potential biomarker for MS.

Previous studies have shown that dysregulated calcium signaling can lead to increased neuronal excitability and inflammation, both of which are central to MS progression [14,16,17].

The significant differences observed in both SARAF levels and expressions suggest that it could serve as a valuable biomarker for MS diagnosis or prognosis, pending further validation through mechanistic and longitudinal studies.

In neonatal platelets, the overexpression of the SARAF protein disrupts calcium homeostasis, which is crucial for maintaining proper platelet function and aggregation. SARAF, a regulator of store-operated calcium entry (SOCE), when overexpressed, leads to impaired calcium signaling, resulting in inadequate clot formation [18]. This disruption is particularly significant in neonatal platelets, which already exhibit unique characteristics and are generally hyporeactive compared to adult platelets. The resulting calcium imbalance not only affects platelet aggregation but also compromises hemostatic functions, increasing the risk of bleeding disorders in newborns.

While SARAF overexpression generally appears to have protective effects against disease progression by modulating calcium entry, its role can vary depending on the cellular context and disease type. For instance, in neonatal platelets, SARAF overexpression leads to impaired calcium homeostasis, which could have adverse effects on platelet function [18]. This highlights the complexity of SARAF’s role in different physiological and pathological conditions.

In models of MS, such as experimental autoimmune encephalomyelitis (EAE), astrocytes exhibit hyperactive calcium transients and dysfunctional responses to neurotransmitters [19]. Aberrant calcium signaling in astrocytes has been linked to impaired synaptic plasticity and cognitive decline in MS [19].

The elevated levels of SARAF in MS patients suggest its involvement in the inflammatory processes’ characteristic of the disease. Previous research has linked dysregulated calcium signaling to increased neuronal excitability and inflammation, which are central to MS progression [14,20]. The significant correlation between SARAF levels and inflammatory markers further supports the hypothesis that SARAF may serve as a biomarker for MS activity [18].

This cytosolic calcium accumulation probably contributes to the inflammatory and neurodegenerative processes in MS. Elevated SARAF levels may drive the hyperactivity of autoreactive T cells, promoting autoimmune responses and cytokine production. Despite increased intracellular calcium, the reduced expression of STIM1 and. suggests a paradoxical dysregulation in calcium signaling, impairing immune cell function, and exacerbating inflammation.

The progressive increase in SARAF levels suggests a temporal role in MS progression. Elevated SARAF levels enhance the activation of STIM1 and ORAI1, increasing calcium influx into the cytosolic space. Simultaneously, we propose that SARAF inhibits the transfer of calcium to the ER via the SERCA channel, preventing STIM1 inactivation. This sustained activation of STIM1 perpetuates calcium accumulation in the cytosolic space. However, at critical SARAF thresholds, a negative feedback mechanism may activate, suppressing the activity and gene expression of STIM1 and ORAI1 (Figure 6). This dynamic regulation highlights SARAF’s complex role in calcium signaling and MS pathophysiology.

Moreover, the observed increase in SARAF expression may have critical implications for T cell function. Given that calcium signaling is essential for T cell activation, the elevation of SARAF could exacerbate the hyperactivity of autoreactive T cells in MS. This aligns with findings from [14], who emphasized the role of SARAF in regulating SOCE, a critical pathway for T cell responses.

The dysregulation of calcium signaling, as evidenced by the elevated intracellular calcium levels and the reduced expression of STIM1 and ORAI1, suggests a complex interplay in the pathophysiology of MS. This dysregulation may contribute not only to T cell hyperactivity but also to neuronal damage and demyelination, hallmark features of the disease [21].

Calcium ions are essential for cellular processes, and their dysregulation is implicated in neurodegenerative conditions, including MS [15,17]. The elevation of SARAF in MS patients raises important questions about its role in disease mechanisms. As SARAF is known to regulate calcium homeostasis, its increased expression may lead to altered calcium influx, potentially exacerbating inflammation and neurodegeneration. This aligns with previous studies suggesting that dysregulated calcium signaling is implicated in MS pathology [17]. The disruption of calcium signaling pathways in neurons can lead to neurodegeneration, highlighting the need for neuroprotective strategies targeting calcium channels [14,22,23,24,25,26].

The Light Cycler analysis corroborated these findings, revealing a fold change in SARAF expression of 3.829 ± 0.04422 in PBMCs from MS patients compared to a baseline of 1 in healthy controls (*p* < 0.0001). This dramatic increase in expression underscores the likelihood that SARAF may influence calcium signaling pathways in T cells, which are known to be dysregulated in autoimmune conditions like MS. Research by [14] has shown that SARAF plays a critical role in regulating store-operated calcium entry (SOCE), a mechanism essential for T cell activation and function.

Given that calcium signaling is crucial for T cell activation, the elevated SARAF levels may contribute to the hyperactivity of autoreactive T cells observed in MS. Studies indicate that altered calcium homeostasis leads to increased T cell proliferation and cytokine production, exacerbating autoimmune responses [20]. The dysregulation of calcium signaling pathways in MS may, therefore, be a key factor driving the pathophysiology of the disease, with SARAF serving as a potential mediator of these effects.

Furthermore, the interplay between SARAF and other calcium-regulating proteins, such as STIM1 and ORAI1, highlights the complexity of calcium signaling in MS. Our findings of reduced expression levels of STIM1 and ORAI1 in MS patients suggest that while SARAF is elevated, the overall calcium signaling machinery may still be impaired, leading to a paradoxical state of increased calcium levels but dysfunctional signaling [17,27,28,29,30,31]. This dysregulation could provide insights into therapeutic strategies targeting calcium signaling pathways to restore balance in immune responses.

The results of this study align with the existing literature that emphasizes the role of calcium homeostasis in autoimmune diseases. Elevated SARAF levels have been observed in other inflammatory conditions, indicating its potential as a universal biomarker for autoimmune activity [14,18]. The findings also resonate with studies highlighting the importance of calcium signaling in T cell function and the implications of its dysregulation in autoimmune disorders [20,32].

In conclusion, this study effectively highlights the potential role of SARAF in the dysregulation of calcium signaling in MS patients. The significant elevation of SARAF levels in MS patients highlights its potential role in the disease’s inflammatory processes and suggests its viability as a biomarker. Further research is essential to elucidate the precise mechanisms by which SARAF contributes to MS and to explore its potential as a therapeutic target in managing this complex autoimmune disorder. The significant differences in calcium levels and the expression of STIM1 and ORAI1 in MS patients reinforce the hypothesis of calcium signaling dysregulation as a contributing factor in the pathophysiology of MS. These findings not only support existing literature but also open new avenues for research and potential therapeutic interventions.

## 4. Materials and Methods

### 4.1. Subjects

The study was approved by the Ethics Committee of the Arabian Gulf University, Manama, Bahrain, and included 45 randomly selected Bahraini MS patients (male and female, aged 15–56 years) under treatment with various drugs (Betaferon, Tysabri, and Gelina). All MS patients were assessed independently by a neurologist and diagnosed clinically and radiologically according to the Modified McDonald criteria 2010 (Dublin 2010). The clinical course was relapse remission (RR); primary progressive (PP); and secondary progressive.

### 4.2. Demographics and Clinical Characteristics

The study populations were selected from Salmaniya medical complex and Blood Bank, Bahrain. It included 45 Bahraini MS patients, 24 males and 21 females (RR 40, PP 3, SP 2). The mean age of the patients with MS was 30.8 years (SD, ±9.1), range 15–56 years. The mean disease duration (DD) of the symptoms was 7 years (SD, ±5.9) range 1–26 years. Evaluation of MS progression and the degree of disability by EDSS score ranged from 1 to 8 (median = 3.5). Patients with MS were under treatment with various drugs, Betaferon, Tysabri, and Gelina. To better understand disease progression, MS patients were divided into two subgroups based on disease duration: Early MS (less than 10 years) and Late MS (more than 10 years).

### 4.3. Blood Sample Collection

Blood samples were collected from MS patients and healthy volunteers into plain tubes and two EDTA tubes for ELISA, RNA purification, and mononuclear cell isolation. Serum samples were allowed to clot overnight at 4 °C before centrifugation for 20 min at approximately 1000× *g*. The serum was then stored at −80 °C until analysis.

### 4.4. Measurement of SARAF Levels by ELISA

To quantify the levels of Store-Operated Calcium Entry-Associated Regulatory Factor (SARAF) in serum samples from both MS patients and healthy controls, the Human Transmembrane Protein 66 (TMEM66/SARAF) ELISA Kit was utilized. This kit employs a double antibody sandwich technique, which allows for the specific detection of (TMEM66/SARAF) in serum, plasma, or cell culture supernatants (MyBioSource, San Diego, CA, USA) Cat No. MBS26027332023). Prior to the assay, the samples were prepared according to the manufacturer’s instructions, ensuring that all reagents were equilibrated to room temperature. SARAF levels were quantified in serum samples using the Human Transmembrane Protein 66 (TMEM66/SARAF) ELISA Kit according to the manufacturer’s instructions. Optical density was measured at 450 nm, and concentrations were determined using a standard curve. The concentration of (TMEM66/SARAF) in the samples was determined by comparing the OD values to a standard curve generated from known concentrations of (TMEM66/SARAF).

### 4.5. RNA Isolation and cDNA Synthesis

Total RNA was extracted from EDTA blood samples using the QIAamp RNA Blood Mini Kit (Qiagen Inc., Germantown, MD, USA) following the manufacturer’s protocol. RNA was quantified using a Nanodrop 1000 spectrophotometer and assessed for integrity by agarose gel electrophoresis. For cDNA synthesis, 1 µg of total RNA was reverse transcribed using the High-Capacity cDNA Reverse Transcription Kit (Thermo Fisher Scientific Inc., Waltham, MA, USA) according to the manufacturer’s instructions. The reaction mixture included 1 µg of RNA, 10 µL of 2× Reverse Transcription Master Mix, and RNase-free water to a final volume of 20 µL. The reverse transcription reaction was conducted in a thermal cycler, with incubation at 25 °C for 10 min to allow for primer annealing, followed by 37 °C for 120 min for reverse transcription, and finally, 85 °C for 5 min to inactivate the reverse transcriptase. The resulting cDNA was stored at −20 °C for subsequent quantitative PCR analysis.

### 4.6. Measurement of Expression of SARAF, STIM1, and ORAI 1 by Real-Time PCR

The expression levels of SARAF, STIM1, and Orai1 were quantified by real-time PCR using a LightCycler 2.0 (Roche, Basel, Switzerland ) with SYBR Green Master Mix (Applied Biosystems, Waltham, MA, USA). Primers (Table 1) and 500 ng cDNA were used in 20 µL reactions. Cycling conditions included 95 °C for 10 min, followed by 45 cycles of 95 °C (10 s), 60 °C (30 s), and 72 °C (30 s). Expression was normalized to GAPDH, and fold changes were calculated using the 2^−ΔΔCt^ method.

### 4.7. Measurement of Intracellular Calcium Levels

Intracellular calcium levels were measured in peripheral blood mononuclear cells (PBMCs) from MS patients and healthy controls using a colorimetric calcium detection assay kit (Abcam, Cambridge, UK). PBMCs were washed with cold phosphate-buffered saline (PBS) and resuspended in calcium assay buffer. The cells were homogenized to lyse them, and the lysate was centrifuged at 4 °C to remove insoluble materials. The supernatant was collected, and a chromogenic reagent was added to both the samples and standards. After incubation in the dark at room temperature for 10 min, optical density was measured at 575 nm using a Synergy HTX multimode microplate reader (Biotek, Winooski, VT, USA). Calcium levels were quantified relative to a standard curve. Measurements were performed in triplicate for each sample, and data were analyzed using the Mann–Whitney U test.

### 4.8. Statistical Analysis

The analysis was performed using the Mann–Whitney U test to assess the differences between the groups. All statistical analyses were conducted using R (version 4.1.1). Data visualization was performed using an in-house R script. The ggplot2 package was utilized to generate the graphical representation of the results.

## 5. Future Directions

Future studies should focus on longitudinal assessments of SARAF levels in MS patients to determine its utility in monitoring disease progression or response to therapy. Additionally, mechanistic studies, such as in vitro knockdown or the overexpression of SARAF in T cells, are planned to elucidate how SARAF elevation contributes to STIM1 and Orai1 downregulation, potentially clarifying its role in MS pathophysiology. Investigating pharmacological agents that can restore normal calcium signaling could provide novel avenues for treatment. Additionally, exploring the mechanistic pathways through which SARAF influences MS pathology could provide deeper insights into therapeutic targets.

The modulation of SARAF levels could potentially influence the efficacy of cytokine-targeted therapies in autoimmune disease treatment. SARAF is involved in calcium signaling, which plays a crucial role in immune cell function and cytokine production. By modulating SARAF, it may be possible to alter cytokine production and thus impact the effectiveness of therapies targeting these cytokines. This approach could complement existing cytokine-targeted therapies, which have shown varying degrees of success in treating autoimmune diseases.

The significant elevation of SARAF levels in MS patients highlights its potential role in the disease’s inflammatory processes and suggests its viability as a biomarker. Further research is essential to elucidate the precise mechanisms by which SARAF contributes to MS and to explore its potential as a therapeutic target in managing this complex autoimmune disorder. The significant differences in calcium levels and the expression of STIM1 and ORAI1 in MS patients reinforce the hypothesis of calcium signaling dysregulation as a contributing factor in the pathophysiology of MS. These findings not only support the existing literature but also open new avenues for research and potential therapeutic interventions.

## Figures and Tables

**Figure 1 ijms-26-04520-f001:**
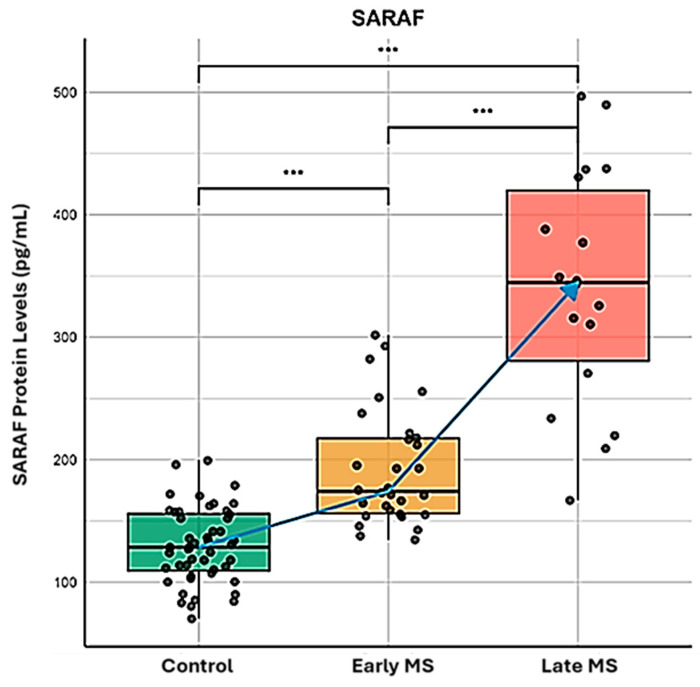
Quantitative analysis of SARAF protein expression during Multiple Sclerosis progression. ELISA-based quantification of SARAF protein concentration (pg/mL) in peripheral blood samples from healthy controls (green box), early-stage MS patients (yellow box), and late-stage MS patients (red box). The data represents three biological replicates per group, analyzed using the Mann–Whitney U test. Blue arrow indicates the progressive elevation of SARAF expression correlating with disease progression. Statistical significance was determined using Mann-Whitney U test (*** *p* < 0.001 for all intergroup comparisons), demonstrating significant upregulation of SARAF protein levels in both early and late MS stages compared to controls, with further significant elevation in late versus early MS.

**Figure 2 ijms-26-04520-f002:**
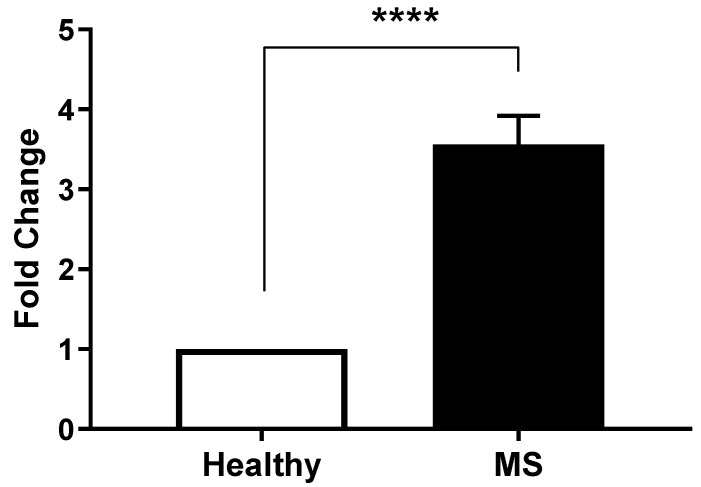
Differential SARAF gene expression in Multiple Sclerosis patients. Quantitative RT-PCR analysis of SARAF mRNA levels in peripheral blood mononuclear cells (PBMCs) isolated from healthy controls and MS patients. Data represents three biological replicates per group. Expression data were normalized to GAPDH as an internal control and presented as fold change relative to healthy controls. Error bars represent mean ± standard error of the mean (SEM). Statistical analysis was performed using unpaired Student’s *t*-test (**** *p* < 0.0001), indicating a highly significant.

**Figure 3 ijms-26-04520-f003:**
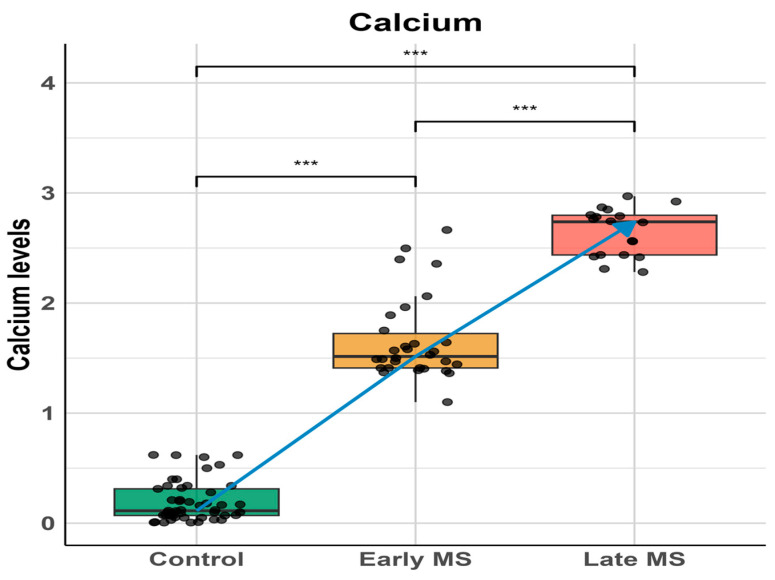
Calcium levels across different stages of Multiple Sclerosis (MS). Box plots show calcium levels in control subjects, early MS, and late MS patients. Individual data points are overlaid to show distribution. A significant progressive increase in calcium levels was observed from control to late MS stage. Statistical significance was determined using one-way ANOVA followed by post-hoc analysis: control vs early MS (*** *p* < 0.001), early MS vs late MS (*** *p* < 0.001), and control vs late MS (*** *p* < 0.001). The blue line represents the trend of increasing calcium levels across disease progression. Data are presented as median with interquartile ranges, and whiskers indicate minimum and maximum values within 1.5 times the interquartile range.

**Figure 4 ijms-26-04520-f004:**
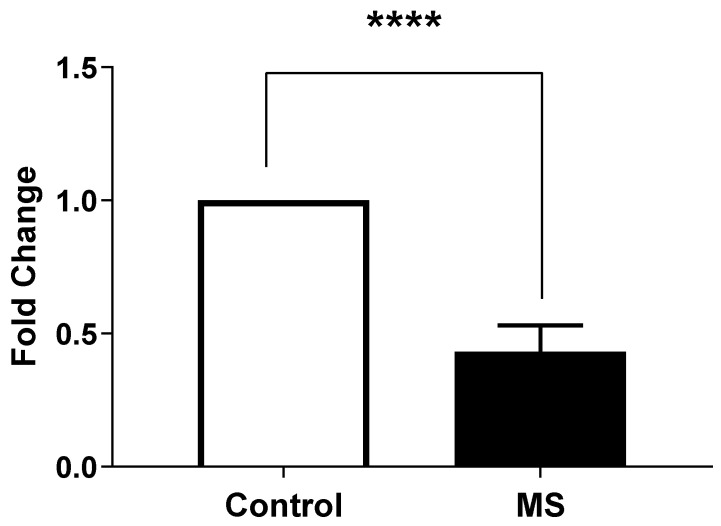
STIM1 gene expression in peripheral blood mononuclear cells (PBMCs) from Multiple Sclerosis (MS) patients compared to healthy controls. Gene expression was measured by quantitative real-time PCR (qRT-PCR) and normalized to GAPDH as housekeeping gene. Data are presented as mean ± SEM from three biological replicates per sample. Statistical analysis was performed using unpaired *t*-test, showing extremely significant difference (**** *p* < 0.0001) between MS patients and healthy controls.

**Figure 5 ijms-26-04520-f005:**
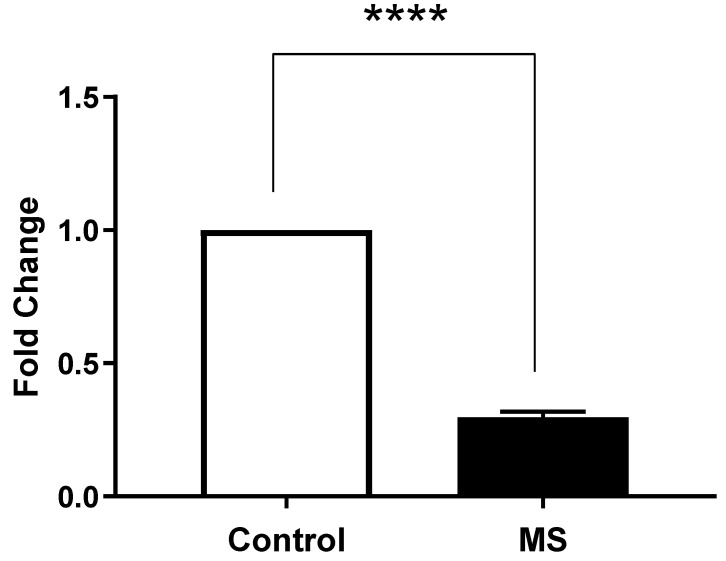
ORAI1 gene expression in peripheral blood mononuclear cells (PBMCs) from Multiple Sclerosis (MS) patients compared to healthy controls. Gene expression was measured by quantitative real-time PCR (qRT-PCR) and normalized to GAPDH as housekeeping gene. Data are presented as mean ± SEM from three biological replicates per sample. Statistical analysis was performed using unpaired *t*-test, showing extremely significant difference (**** *p* < 0.0001) between MS patients and healthy controls.

**Figure 6 ijms-26-04520-f006:**
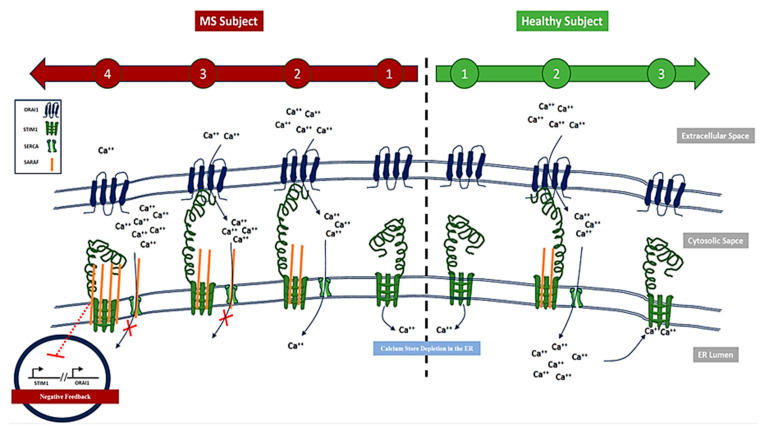
Schematic representation of SARAF protein accumulation in Multiple Sclerosis (MS) patients compared to healthy controls. The left panel illustrates elevated levels of SARAF protein observed in MS patients, while right panel depicts normal SARAF expression in healthy controls. This accumulation may contribute to accumulation of calcium at cytosolic space.

**Table 1 ijms-26-04520-t001:** Primers sequences.

Primer	Accession Number	Forward Primer	Reverse Primer	Product Length
GAPDH	NM_002046.7	TCCCTGAGCTGAACGGGAAG	GGAGGAGTGGGTGTCGCTGT	217 bp
SARAF	NM_016127.6	TGGAACGACCCTGACAGAATG	ACCCATCCCAGCCTTTGTTC	181 bp
Orai 1	NM_032790.4	AGTCGTGGTCAGCGTCCAGCT	AGGTGATGAGCCTCAACGAGCA	159 bp
STIM1	NM_003156.4	TTGACAAGCCGGGTATCTCTG	CATCTGAGGAGGTTTGGGGG	369 bp

## Data Availability

The data presented in this study are available on request from the corresponding author due to privacy and ethical reasons.

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
