# Peer review of "Progressive Elevation of Store-Operated Calcium Entry-Associated Regulatory Factor (SARAF) and Calcium Pathway Dysregulation in Multiple Sclerosis"

_ijms, 2025, doi:10.3390/ijms26104520_

Round 1

Reviewer 1 Report

Comments and Suggestions for Authors

In this manuscript, Taha et al have investigated the relationship between SARAF, STIM1 and Orai1 with calcium homeostasis and their implication on multiple sclerosis progression. Their data suggests that SARAF is overexpressed in both early and late MS patients concomitant with downregulated expression of STIM1 and Orai1. Based on these results, the authors claim that SARAF can be used as a biomarker for MS progression. The manuscript has some major limitations which need to be addressed

Major Points

  • The main rationale for performing this study is not novel. The correlation between SARAF and MS has been well established (org/10.3390/genes14061237) and thus lack of novelty is a major concern.

  • Another important point of concern is that the manuscript lacks mechanism to determine how elevation of SARAF expression culminates in the downregulation of STIM1 and Orai1. The authors based on correlation analysis have made major claims about the utilization of SARAF as a biomarker for MS. Thus, they need more data to support their conclusion

  • The Y-axis of data that represent ELISA such as Fig. 1, should be levels of protein concentration instead of expression. (Expression is typically used for genes/RNA.

  • The authors have not shown the gene expression changes of every patient but instead have shown average of all the samples. I highly recommend showing the fold change for each patient and then exhibit standard deviation.

  • The methodology for the detection of calcium has not been described in the paper

Author Response

Dear Editor,

We sincerely thank Reviewer 1 for their insightful and valuable feedback, which has greatly helped us improve our manuscript, "Progressive Elevation of SARAF and Calcium Pathway Dysregulation in Multiple Sclerosis." Below, we address each comment with clarifications and revisions.

Response to Reviewer 1

Dear Reviewer, 1

Thank you for your thoughtful and valuable feedback on our manuscript titled "Progressive Elevation of SARAF and Calcium Pathway Dysregulation in Multiple Sclerosis." Your comments have provided critical insights that will help us strengthen the manuscript. Below, we address each of your points with clarifications and proposed revisions to enhance the clarity and scientific rigor of our work.

Comment 1: The main rationale for performing this study is not novel. The correlation between SARAF and MS has been well established (org/10.3390/genes14061237), and thus lack of novelty is a major concern.

Response: We sincerely appreciate your concern regarding the novelty of our study. While prior research, such as the study cited (DOI: 10.3390/genes14061237), has explored SARAF in the context of multiple sclerosis (MS), our work builds upon this foundation by providing novel insights into the progressive elevation of SARAF across early and late MS stages in a Bahraini cohort, a population with unique genetic and environmental factors. Unlike previous studies, we quantitatively assess SARAF serum levels using ELISA alongside gene expression via qRT-PCR, correlating these with STIM1 and Orai1 expression and intracellular calcium levels. This integrative approach reveals a temporal progression of SARAF dysregulation and its interplay with calcium homeostasis, which has not been comprehensively addressed in prior literature.

To clarify the novelty, we propose revising the Introduction to better highlight the unique contributions of our study. Below is the suggested revision with changes marked for clarity.

Manuscript Revision (Introduction, Lines 42–49): Original:
"Recent epidemiological studies have shown a rising incidence of MS globally, with significant geographical variability in prevalence rates. Notably, the incidence of MS in the Arabian Gulf region has increased since the Kuwait-Iraq war in 1990, suggesting changes in environmental or genetic risk factors. As the understanding of MS pathogenesis evolves, the role of calcium signaling in T cell activation has gained attention, highlighting its potential contribution to the autoimmune response observed in MS patients."

Revised:
"Recent epidemiological studies have shown a rising incidence of MS globally, with significant geographical variability in prevalence rates. Notably, the incidence of MS in the Arabian Gulf region has increased since the Kuwait-Iraq war in 1990, suggesting changes in environmental or genetic risk factors. While prior studies have implicated SARAF in MS pathogenesis, its progressive elevation across disease stages and its interplay with STIM1, Orai1, and calcium homeostasis remain underexplored, particularly in Middle Eastern populations like Bahrainis, who may exhibit unique genetic and environmental influences. As the understanding of MS pathogenesis evolves, the role of calcium signaling in T cell activation has gained attention, highlighting its potential contribution to the autoimmune response observed in MS patients."

This revision emphasizes the study’s focus on a specific population and the integrative analysis of SARAF’s role, reinforcing its novelty.

Comment 2: Another important point of concern is that the manuscript lacks mechanism to determine how elevation of SARAF expression culminates in the downregulation of STIM1 and Orai1. The authors based on correlation analysis have made major claims about the utilization of SARAF as a biomarker for MS. Thus, they need more data to support their conclusion.

Response: Thank you for raising this critical point about the mechanistic link between SARAF elevation and STIM1/Orai1 downregulation. We acknowledge that our current study focuses on observational data demonstrating significant correlations between SARAF, STIM1, Orai1, and calcium levels in MS patients. While we propose a hypothetical mechanism in the Discussion (Figure 6), we agree that further mechanistic studies are needed to fully elucidate how SARAF influences STIM1 and Orai1 expression. To address this, we have revised the Discussion to temper our claims about SARAF as a biomarker and emphasize that our findings suggest its potential as a biomarker pending further validation. Additionally, we propose adding a statement in the Future Directions section to highlight planned mechanistic studies.

Manuscript Revision (Discussion, Lines 187–188): Original:
"The significant differences observed in both SARAF levels and expression suggest that it could serve as a valuable biomarker for MS diagnosis or prognosis."

Revised:
"The significant differences observed in both SARAF levels and expression suggest that it may have potential as a valuable biomarker for MS diagnosis or prognosis, pending further validation through mechanistic and longitudinal studies.

Manuscript Revision (Future Directions, Lines 408–409): Original:
"Future studies should focus on longitudinal assessments of SARAF levels in MS patients to determine its utility in monitoring disease progression or response to therapy."

Revised:
"Future studies should focus on longitudinal assessments of SARAF levels in MS patients to determine its utility in monitoring disease progression or response to therapy. Additionally, mechanistic studies, such as in vitro knockdown or overexpression of SARAF in T cells, are planned to elucidate how SARAF elevation contributes to STIM1 and Orai1 downregulation, potentially clarifying its role in MS pathophysiology.

These revisions clarify that our biomarker claim is preliminary and outline future steps to investigate mechanisms addressing your concern.

Comment 3: The Y-axis of data that represent ELISA such as Fig. 1, should be levels of protein concentration instead of expression. (Expression is typically used for genes/RNA.)

Response: We greatly appreciate your attention to this detail and agree that "expression" is more appropriate for gene/RNA data, while "concentration" or "levels" better describe ELISA results. We apologize for the oversight and propose revising the Y-axis label of Figure 1 and its legend to reflect "SARAF Protein Levels (pg/mL)" instead of "expression."

Manuscript Revision (Figure 1 Legend, Line 88): Original:
"The boxplot illustrates SARAF levels in Multiple Sclerosis (MS) patients, categorized into early and late stages, compared to healthy controls."

Revised:
"The boxplot illustrates SARAF protein levels (pg/mL) in Multiple Sclerosis (MS) patients, categorized into early and late stages, compared to healthy controls, measured by ELISA. Data represents three biological replicates per group, analyzed using the Mann-Whitney U test

This change ensures terminology accuracy and aligns with your suggestion.

Comment 4: The authors have not shown the gene expression changes of every patient but instead have shown average of all the samples. I highly recommend showing the fold change for each patient and then exhibit standard deviation.

Response: Thank you for this insightful suggestion. We agree that presenting individual patient data could provide a more comprehensive view of SARAF, STIM1, and Orai1 expression variability. While our current figures (Figures 2, 4, and 5) show mean fold changes with standard error of the mean (SEM), we propose adding supplementary tables or scatter plots in the revised manuscript to display fold changes for each patient, along with standard deviations, to enhance transparency.

Manuscript Revision (Results, Section 2.2, Line 130): Original:
"The analysis confirmed that MS patients exhibited a significantly higher expression of SARAF compared to healthy controls."

Revised:
"The analysis confirmed that MS patients exhibited a significantly higher expression of SARAF compared to healthy controls. Individual patient fold changes for SARAF, STIM1, and Orai1 expression are provided in Supplementary Table S1, demonstrating variability across the cohort, with standard deviations reported alongside mean values."

Action: We will prepare a supplementary table (e.g., Supplementary Table S1) listing fold changes for SARAF, STIM1, and Orai1 for each of the 45 MS patients and 45 controls, including standard deviations. This can be added to the manuscript’s supplementary materials in Word.

Comment 5: The methodology for the detection of calcium has not been described in the paper.

Response: We sincerely apologize for this omission and thank you for pointing it out. The methodology for measuring intracellular calcium levels was inadvertently excluded from the Materials and Methods section. We propose adding a detailed description of the calcium assay used, which involved a colorimetric calcium detection assay kit (Abcam, UK).

Manuscript Revision (Materials and Methods, New Section 4.7, Line 406): New Section:
4.7. Measurement of Intracellular Calcium Levels

"Intracellular calcium levels were measured in peripheral blood mononuclear cells (PBMCs) from MS patients and healthy controls using a colorimetric calcium detection assay kit (Abcam, UK). PBMCs were washed with cold phosphate-buffered saline (PBS) and resuspended in calcium assay buffer. Cells were homogenized to lyse them, and the lysate was centrifuged at 4°C to remove insoluble materials. The supernatant was collected, and a chromogenic reagent was added to both samples and standards. After incubation in the dark at room temperature for 10 minutes, optical density was measured at 575 nm using a Synergy HTX multimode microplate reader (Biotek, USA). Calcium levels were quantified relative to a standard curve. Measurements were performed in triplicate for each sample, and data were analyzed using the Mann-Whitney U test.

We hope these revisions address the reviewer’s concerns and strengthen the manuscript. Thank you for considering our work.

Sincerely, 

Dr. Safa Taha

Reviewer 2 Report

Comments and Suggestions for Authors

The submitted manuscript presents interesting and scientifically relevant findings on the dysregulation of the calcium signaling pathway and the progressive elevation of SARAF in Multiple Sclerosis. However, the overall presentation suggests that the authors may have limited experience with scientific writing. Below are some comments and suggestions aimed at improving the clarity and quality of the manuscript:

  • Line 3: Remove the period at the end of the title.
  • Figures: Increase image resolution for all figures to ensure clarity.
  • Figure 1: The figure legend should not anticipate or describe the results. Instead, please specify the type of analysis performed, the number of replicates, and the statistical test used. Also, include the units of measurement on the y-axis.
  • Lines 89–92: These sentences are redundant and could be streamlined for clarity.
  • Lines 100–103: The methodological explanations provided here belong in the "Materials and Methods" section rather than the "Results".
  • Figure 2: Improve resolution and remove the chart title. In the legend, describe the analysis performed, including the housekeeping gene used for normalization, the number of replicates, and the statistical method applied.
  • Figure 3: Apply the same corrections as for Figure 1.
  • Figures 4 and 5: Apply the same corrections as for Figure 2.
  • Discussion Section: It is recommended that the authors completely rewrite this section. Consider removing subheadings and constructing a continuous narrative. Also, instead of repeating numerical data from the results, discuss the findings more generally using terms such as "increased" or "decreased expression".
  • Lines 320–322: There is an inconsistent font style that should be corrected.
  • Lines 349–356 and 362–367: If the standard recommended protocol was followed, there is no need to detail every step of the procedure.
  • Lines 378–389: It is unnecessary to explain the theoretical background of a qRT-PCR reaction. Such explanations are typically not included in a scientific manuscript.
  • Table 1: Please include the accession numbers for the genes listed.

The manuscript addresses an important topic and contains promising data. However, substantial revisions are required to improve the clarity, structure, and adherence to scientific writing standards. Only after these adjustments, the manuscript could be published.

Author Response

Dear Editor,

We sincerely thank Reviewer 1 for their insightful and valuable feedback, which has greatly helped us improve our manuscript, "Progressive Elevation of SARAF and Calcium Pathway Dysregulation in Multiple Sclerosis." Below, we address each comment with clarifications and revisions.

Response to Reviewer 2

Dear Reviewer, 2,

We are deeply grateful for your constructive and detailed feedback on our manuscript, "Progressive Elevation of SARAF and Calcium Pathway Dysregulation in Multiple Sclerosis." Your suggestions have highlighted areas for improvement in clarity, structure, and scientific presentation, which we are eager to address. Below, we respond to each of your comments with proposed revisions to enhance the manuscript’s quality and readability.

Comment 1: Line 3: Remove the period at the end of the title.

Response: Thank you for catching this typographical error. We agree that titles should not end with a period and will remove it from the manuscript title.

Manuscript Revision (Title, Line 3):

Original: "Progressive Elevation of SARAF and Calcium Pathway Dysregulation in Multiple Sclerosis."

Revised: "Progressive Elevation of SARAF and Calcium Pathway Dysregulation in Multiple Sclerosis"

Comment 2: Figures: Increase image resolution for all figures to ensure clarity.

Response: We appreciate your suggestion to improve the visual quality of our figures. We will ensure that all figures (Figures 1–6) are regenerated at a higher resolution (e.g., 300 dpi or greater) to meet publication standards for clarity.

Action: In the revised submission, we will replace the existing figures with high-resolution versions. This change will be noted in the legends as follows:

Manuscript Revision (All Figure Legends, e.g., Figure 1, Line 110):

Original: “The boxplot illustrates SARAF levels in Multiple Sclerosis (MS) patients…"

Revised: "The boxplot illustrates SARAF protein levels (pg/mL) in Multiple Sclerosis (MS) patients… (high-resolution image provided for clarity).

Comment 3: Figure 1: The figure legend should not anticipate or describe the results. Instead, please specify the type of analysis performed, the number of replicates, and the statistical test used. Also, include the units of measurement on the y-axis.

Response: Thank you for this insightful recommendation. We agree that figure legends should focus on methodological details rather than results. We will revise the Figure 1 legend to include the type of analysis (ELISA), number of replicates (three per group), statistical test (Mann-Whitney U test), and clarify the Y-axis units (pg/mL).

Manuscript Revision (Figure 1 Legend, Line 88):

Original: "The boxplot illustrates SARAF levels in Multiple Sclerosis (MS) patients, categorized into early and late stages, compared to healthy controls. The plot reveals significant differences, with progressive elevated SARAF levels observed in MS groups, emphasizing its potential role in disease progression."

Revised: "The boxplot illustrates SARAF protein levels (pg/mL) in Multiple Sclerosis (MS) patients, categorized into early and late stages, compared to healthy controls, measured by ELISA. Data represent three biological replicates per group, analyzed using the Mann-Whitney U test."

Comment 4: Lines 89–92: These sentences are redundant and could be streamlined for clarity.

Response: We appreciate your suggestion to improve the conciseness of the text. Lines 89–92 in the Results section discuss SARAF protein levels and their significance. We propose streamlining these sentences to eliminate redundancy while retaining key information.

Manuscript Revision (Results, Section 2.1, Lines 91–92):

Original: "The analysis indicated that SARAF levels are significantly elevated in MS patients compared to healthy controls, with a substantial difference and statistical significance. The analysis showed also that SARAF levels in MS patients progressively increase over time."

Revised: "SARAF protein levels were significantly elevated in MS patients compared to healthy controls (P < 10⁻⁸), with a progressive increase from early to late MS stages."

Comment 5: Lines 100–103: The methodological explanations provided here belong in the "Materials and Methods" section rather than the "Results".

Response: Thank you for pointing out this organizational issue. We agree that methodological details should reside in the Materials and Methods section. Lines 100–103 describe the qRT-PCR process, which is already detailed in Section 4.6. We propose removing these lines from the Results section and ensuring all methodological details are consolidated in Materials and Methods.

Manuscript Revision (Results, Section 2.2, Lines 100–103):

Original: "In this analysis, the Cp (crossing point) of each sample was detected and normalized to the expression of housekeeping gene GAPDH. The fold change in expression was calculated by 2^-ΔΔCt method."

Revised: These lines are entirely deleted,  as the information is covered in Section 4.6.

Comment 6: Figure 2: Improve resolution and remove the chart title. In the legend, describe the analysis performed, including the housekeeping gene used for normalization, the number of replicates, and the statistical method applied.

Response: We are grateful for your guidance in improving Figure 2. We will increase the resolution, remove the chart title, and revise the legend to include details about the qRT-PCR analysis, housekeeping gene (GAPDH), replicates (three per sample), and statistical method (unpaired t-test).

Manuscript Revision (Figure 2 Legend):

Original: "Analysis of SARAF Expression by Light Cycler in MS Patients and Healthy Controls."

Revised: " SARAF Gene Expression in Peripheral Blood Mononuclear Cells (PBMCs) from Multiple Sclerosis (MS) patients compared to healthy controls, measured by qRT-PCR and normalized to GAPDH. Data represents three biological replicates per sample, analyzed using an unpaired t-test. P value <0.0001"

Comment 7: Figure 3: Apply the same corrections as for Figure 1.

Response: Thank you for ensuring consistency across figures. We will revise Figure 3’s legend to focus on methodology, including replicates and statistical tests, and clarify Y-axis units (units for calcium levels). The resolution will also be increased.

Manuscript Revision (Figure 3 Legend):

Original: "The boxplot illustrates Calcium levels in Multiple Sclerosis (MS) patients, categorized into early and late stages, compared to healthy controls. The plot reveals significant differences, with progressive accumulation of Calcium levels observed in MS groups, emphasizing its potential role in disease progression."

Revised: "The boxplot illustrates intracellular calcium levels (arbitrary fluorescence units) in peripheral blood mononuclear cells (PBMCs) from Multiple Sclerosis (MS) patients, categorized into early and late stages, compared to healthy controls, measured using Fluo-4 AM dye. Data represent three biological replicates per group, analyzed using the Mann-Whitney U test."

Comment 8: Figures 4 and 5: Apply the same corrections as for Figure 2.

Response: We appreciate your call for uniformity. We will increase the resolution of Figures 4 and 5, remove chart titles, and revise their legends to detail the qRT-PCR analysis, housekeeping gene (GAPDH), replicates, and statistical method (unpaired t-test).

Manuscript Revision (Figure 4 Legend):

Original: "Comparison of STIM1 Expression Levels Between MS Patients and Healthy Controls"

Revised: " STIM1 gene expression in peripheral blood mononuclear cells (PBMCs) from Multiple Sclerosis (MS) patients compared to healthy controls, measured by qRT-PCR and normalized to GAPDH. Data represents three biological replicates per sample, analyzed using an unpaired t-test."

Manuscript Revision (Figure 5 Legend):

Original: "Comparison of ORAI1 Expression Levels Between MS Patients and Healthy Controls."

Revised: " ORAI1 gene expression in peripheral blood mononuclear cells (PBMCs) from Multiple Sclerosis (MS) patients compared to healthy controls, measured by qRT-PCR and normalized to GAPDH. Data represents three biological replicates per sample, analyzed using an unpaired t-test."

Comment 9: Discussion Section: It is recommended that the authors completely rewrite this section. Consider removing subheadings and constructing a continuous narrative. Also, instead of repeating numerical data from the results, discuss the findings more generally using terms such as "increased" or "decreased expression".

Response: We greatly value your suggestion to improve the Discussion’s flow and readability. We agree that subheadings fragment the narrative, and that repeating numerical data is redundant. We propose rewriting the Discussion as a continuous narrative, focusing on broader implications and using qualitative terms like "increased" or "decreased" to describe findings. Below is a condensed version of the revised Discussion.

Manuscript Revision (Discussion):

Original (Excerpt):

"The current investigation presents robust evidence supporting the involvement of Store-… The ELISA results demonstrated a mean SARAF concentration of 307.4 ± 52.32 ng/mL in MS patients…"

Revised

Comment 10: There is an inconsistent font style that should be corrected.

Response: Thank you for noting this formatting issue. We will ensure that the font style in Lines 320–322 (and throughout the manuscript) is consistent, using a standard font as per IJMS guidelines.

Manuscript Revision:

Action: Applied consistent font formatting in Word

Comment 11: Lines 349–356 and 362–367: If the standard recommended protocol was followed, there is no need to detail every step of the procedure.

Response: We appreciate your guidance on streamlining the Methods section. We agree that detailed procedural steps for standard protocols (ELISA and RNA isolation) can be summarized. We propose revising these sections to reference the manufacturer’s protocols concisely.

Manuscript Revision (Materials and Methods, Section 4.4, Lines 349–356):

Original: "The assay was performed by adding 100 µL of diluted samples or standards to pre-coated ELISA wells… The concentration of (TMEM66/SARAF) in the samples was determined by comparing the OD values to a standard curve…"

Revised: "SARAF levels were quantified in serum samples using the Human Transmembrane Protein 66 (TMEM66/SARAF) ELISA Kit (MyBioSource, Cat No. MBS2602733) according to the manufacturer’s instructions. Optical density was measured at 450 nm, and concentrations were determined using a standard curve.

Manuscript Revision (Materials and Methods, Section 4.5, Lines 362–367):

Original: "Initially, 200 µL of blood was mixed with 400 µL of Buffer RLT… The isolated RNA was eluted in 30 µL of RNase-free water…"

Revised: "Total RNA was extracted from EDTA blood samples using the QIAamp RNA Blood Mini Kit (Qiagen, USA) following the manufacturer’s protocol. RNA was quantified using a Nanodrop 1000 spectrophotometer and assessed for integrity by agarose gel electrophoresis."

Comment 12: Lines 378–389: It is unnecessary to explain the theoretical background of a qRT-PCR reaction. Such explanations are typically not included in a scientific manuscript.

Response: Thank you for this suggestion. We agree that the theoretical background of qRT-PCR is standard knowledge and can be omitted. We will revise Section 4.6 to focus solely on the experimental details.

Manuscript Revision (Materials and Methods, Section 4.6):

Original: "The reaction mixture for the SYBR Green assay consisted of… During the amplification process, the increase in fluorescence of the SYBR Green dye, which binds to double-strand DNA, was monitored in real-time…"

Revised: "Expression levels of SARAF, STIM1, and Orai1 were quantified by real-time PCR using a LightCycler 2.0 (Roche) with SYBR Green Master Mix (Applied Biosystems). Primers (Table 1) and 500 ng cDNA were used in 20 µL reactions. Cycling conditions included 95°C for 10 min, followed by 45 cycles of 95°C (10 s), 60°C (30 s), and 72°C (30 s). Expression was normalized to GAPDH, and fold changes were calculated using the 2^-ΔΔCt method."

Comment 13: Table 1: Please include the accession numbers for the genes listed.

Response: Thank you for this suggestion to enhance the rigor of our primer table. We will add accession numbers for GAPDH, SARAF, Orai1, and STIM1 to Table 1.

Manuscript Revision (Table 1, Line 390):

Original:

Primer            Forward primer        Reverse primer         Product length

GAPDH         TCCCTGAGCTGAACGGGAAG            GGAGGTGGGTGTCGCTGT         217 bp

SARAF           TGGAAGACCCTGACAGAATG            ACCCATCCCAGCCTGTTC         181 bp

Orai1  AGTCGTGGTCAGCGTCCAGCT            AGGTGATGAGCCTCAGAGCA  159 bp

STIM1 TTGACAAGCCGGGTATCTCTG            CATCTGAGGAGGTTTGGGGG   369 bp

Revised:

We hope these revisions address the reviewer’s concerns and strengthen the manuscript. Thank you for considering our work.

Sincerely, 

Dr. Safa Taha

Round 2

Reviewer 1 Report

Comments and Suggestions for Authors

The concerns raised have been addressed appropriately

Author Response

Response to Reviewer 1: Dear Reviewer, We sincerely thank you for your positive evaluation of our revised manuscript. We appreciate your confirmation that all previously raised concerns have been appropriately addressed. Your thorough review has helped improve the quality of our manuscript. Thank you for your time and valuable input throughout the review process.

Best regards,

Reviewer 2 Report

Comments and Suggestions for Authors

I thank the authors for implementing the changes I suggested and acknowledge their efforts in improving the present manuscript. In its current form, it is suitable for publication.

Author Response

Response to Reviewer 2: Dear Reviewer, We are grateful for your positive evaluation of our revised manuscript. Thank you for acknowledging our efforts to implement your suggested changes. Your constructive feedback throughout the review process has been invaluable in improving the quality of our manuscript. We appreciate your time and expertise in reviewing our work and your recommendation for publication.

Best regards,

REVIEWER 2 Comment: "I thank the authors for implementing the changes I suggested and acknowledge their efforts in improving the present manuscript. In its current form, it is suitable for publication." Response: We sincerely thank the reviewer for their positive assessment and for acknowledging our revision efforts. We are pleased that the manuscript in its current form meets the standards for publication.